# Beckwith–Wiedemann Syndrome in Newborn of Mother with HELLP Syndrome/Preeclampsia: An Analysis of Literature and Case Report with Fetal Growth Restriction and Absence of *CDKN1C* Typical Pathogenic Genetic Variation

**DOI:** 10.3390/ijms241713360

**Published:** 2023-08-29

**Authors:** Jakub Staniczek, Maisa Manasar-Dyrbuś, Agnieszka Drosdzol-Cop, Rafał Stojko

**Affiliations:** Chair and Department of Gynecology, Obstetrics and Gynecologic Oncology, Medical University of Silesia, 40-211 Katowice, Poland

**Keywords:** Beckwith–Wiedemann syndrome, HELLP syndrome, preeclampsia, *CDKN1C*, fetal growth restriction

## Abstract

Beckwith–Wiedemann Syndrome (BWS) is an imprinting disorder, which manifests by overgrowth and predisposition to embryonal tumors. The evidence on the relationship between maternal complications such as HELLP (hemolysis, elevated liver enzymes, and low platelet count) and preeclampsia and the development of BWS in offspring is scarce. A comprehensive clinical evaluation, with genetic testing focused on screening for mutations in the CDKN1C gene, which is commonly associated with BWS, was conducted in a newborn diagnosed with BWS born to a mother with a history of preeclampsia and HELLP syndrome. The case study revealed typical clinical manifestations of BWS in the newborn, including hemihyperplasia, macroglossia, midfacial hypoplasia, omphalocele, and hypoglycemia. Surprisingly, the infant also exhibited fetal growth restriction, a finding less commonly observed in BWS cases. Genetic analysis, however, showed no mutations in the *CDKN1C* gene, which contrasts with the majority of BWS cases. This case report highlights the complex nature of BWS and its potential association with maternal complications such as preeclampsia and HELLP syndrome. The atypical presence of fetal growth restriction in the newborn and the absence of *CDKN1C* gene mutations have not been reported to date in BWS.

## 1. Introduction

Beckwith–Wiedemann Syndrome is a rare and clinically diverse genetic disorder characterized by somatic overgrowth and an increased risk of embryonal tumors. BWS affects 1 in 10,500 to 13,700 newborns worldwide [1]. First described independently by J. Bruce Beckwith and Hans-Rudolf Wiedemann in the 1960s, BWS is primarily associated with alterations in the imprinted region on chromosome 11p15.5 [1]. Patients with BWS exhibit a wide spectrum of clinical features, including macrosomia, macroglossia, abdominal wall defects, visceromegaly, and neonatal hypoglycemia, among others. The etiology of BWS is pleiotropic, involving the dysregulation of imprinted genes implicated in cell growth and proliferation. In approximately 85% of cases, BWS arises due to epigenetic changes such as loss of methylation at the imprinting control region 2 *ICR2* or gain of methylation at *ICR1* within the 11p15.5 region. These epigenetic changes lead to the altered expression of genes involved in cell cycle regulation, with the most affected gene being *CDKN1C* (cyclin-dependent kinase inhibitor 1C), a negative regulator of cell division [1,2,3,4,5,6,7,8].

Recent research has also explored the potential associations between BWS and adverse maternal conditions during pregnancy, such as preeclampsia and HELLP (hemolysis, elevated liver enzymes, low platelet count) syndrome—a severe variant of preeclampsia. Preeclampsia is a pregnancy-specific hypertensive disorder that affects 2–8% of pregnancies worldwide. Although the pathophysiology of these maternal complications remains incompletely understood, there is evidence suggesting that they might influence the expression of imprinted genes and contribute to the development of BWS in offspring [9,10].

This case report presents a rare and intriguing patient with BWS without typical pathogenic genetic variation, in an infant born to a mother with preeclampsia and HELLP syndrome.

## 2. Case Presentation

### 2.1. Patient Information and Prenatal Findings

A 33-year-old pregnant woman, at 36 and 2/7 weeks’ gestation, in her first pregnancy, with no significant medical history was admitted to the Pregnancy Pathology Ward for fetal well-being assessment and induction of labor. The patient was diagnosed with grade 1 Fetal Growth Restriction (FGR) based on the recommendations of the International Society of Ultrasound in Obstetrics and Gynecology (ISUOG) and the Polish Society of Gynecologists and Obstetricians [11,12,13,14]. On the day of admission, fetal well-being was assessed with the following results: estimated fetal weight 0 percentile, cerebral–placental ratio 1.77 (26th percentile), and mean pulsatility index in the uterine artery of 0.65 (40th percentile). No abnormalities were reported in the laboratory tests, which are summarized in Table 1. Neither the patient nor her husband had any history of genetic disorders in their families. From the 30th week of pregnancy, the patient took methyldopa (2 × 250 mg) to manage pregnancy-induced hypertension (PIH), which was optimally controlled.

Throughout her pregnancy, the patient did not report any complaints or complications, and it progressed without any issues. She underwent six standard pregnancy ultrasound examinations, as per the Standards of the Polish Society of Gynecologists and Obstetricians, and four additional ultrasound examinations along with fetal echocardiography at a tertiary center for prenatal diagnostics. The first-trimester prenatal examination revealed normal fetal anatomy and a low risk of trisomy, preeclampsia, and preterm delivery. The combined test values were as follows: free subunit of β-hCG 0.87 MoM, PAPP-A 0.701 MoM, and AFP 7.12 IU/l. All ultrasound examinations showed no structural defects in the fetus. There were no ultrasonographic features of placental mesenchymal dysplasia or other identifiable placental pathologies. The fetus exhibited slow weight gain starting from the 20th week of gestation, assessed using Hadlock’s formula 3 and The World Health Organization Fetal Growth Charts [15,16,17]. Grade 1 FGR was diagnosed in the 30th week of pregnancy, in accordance with the recommendations [13,14]. Following the diagnosis of FGR, weekly assessments of the cerebroplacental index and amniotic fluid volume, using the largest fluid pocket, were performed. Additionally, from the 34th week of pregnancy, a 40–60 min cardiotocographic record with short-term variation assessment was conducted once a week. The ultrasonographic measurement results are presented in Table 1.

### 2.2. Therapeutic Interventions

At 36 + 3 weeks of gestation, the patient started labor pre-induction. Before the pre-induction of labor, fetal well-being was determined by CTG and Doppler ultrasonography and biophysical profile. Labor pre-induction involved the use of a Foley catheter, which was removed after 24 h. Subsequently, an insert containing dinoprostone was utilized, and it was removed after 8 h due to the onset of regular contractile actions of the uterus. A permanent cardiotocographic record was then connected and Group B streptococcal prophylaxis was started. The amniotic fluid drained 20 min before delivery and the patient gave birth to a male newborn weighing 1830 g (<3%), with dimensions of 47 cm in length (50%) and a head circumference of 30 cm (<3%) (percentiles according to Fenton preterm growth charts [18]). He received 6, 8, 9, and 10 points on the Apgar scale at 1, 3, 5, and 10 min, respectively, and the umbilical blood was pH 7.1.

#### 2.2.1. Therapeutic Interventions in Mother

Within 20 h after delivery, the patient reported epigastric pain and dyspnea. On physical examination, she had an average blood pressure of 180/100 mmHg in several measurements. The dose of methyldopa was increased to 3 × 500 mg and nifedipine 2 × 20 mg was added, obtaining normal blood pressure.

However, within the next 2 h, her blood pressure increased again to 190/100 mmHg. As a temporary measure, urapidil 12.5 mg was administered intravenously, 2 g MgSO_4_ in an infusion, and a third antihypertensive drug, metoprolol 2 × 50 mg, was added. The control laboratory test showed a reduced platelet count, elevated liver parameters, and symptoms of hemolysis, leading to the diagnosis of Complete HELLP Syndrome (Tennessee System) and Class 1 HELLP Syndrome (Mississippi System). The patient received standard treatment, including steroid therapy with dexamethasone 2 × 8 mg, ursodeoxycholic acid 2 × 500 mg, low-molecular-weight heparin 1 × 0.6 (next 2 × 0.6), and platelet concentrate transfusion. Eventually, an improvement in the laboratory results and general condition was observed. The laboratory results of the patient (mother) before and after delivery are included in Table 2.

#### 2.2.2. Therapeutic Interventions in Newborn

After birth, the newborn required assistance in lung expansion with the use of positive end-expiratory pressure (CPAP), but did not need oxygen therapy. On the first postnatal day, the newborn exhibited a tendency toward hypoglycemia, despite receiving intravenous infusions with 10% glucose. Elevated TSH values (16,850 µIU/mL) were found, along with normal fT4 values (2.13 ng/dL). After consultation with the pediatric endocrinologist, levothyroxine 1 × 6.25 mg was initiated.

The newborn showed features of facial dysmorphism—hypertelorism and macroglossia. The newborn did not display any signs of congenital infection, and laboratory tests ruled out any signs of such infection. The newborn also underwent a consultation with a pediatric cardiologist, who detected the presence of an open foramen ovale, which was considered normal for the baby’s age. Furthermore, the newborn underwent both a transfontanelle ultrasound and an abdominal ultrasound, which did not reveal any pathology. Due to severe jaundice, the newborn required phototherapy.

The newborn was discharged on the 11th day of life, after reaching 2000 g and vaccination against tuberculosis. Outpatient control at the Pediatric Endocrinology Clinic and Clinical Genetics was recommended.

### 2.3. Genetic Testing and Follow-up

Upon examination, prominent macroglossia and hypertelorism were evident. By the third week of life, signs of hemihyperplasia, midfacial hypoplasia, and a prominent occiput became apparent. The newborn was referred to a reference Department of Medical Genetics for children. A follow-up abdominal ultrasound revealed the presence of a small omphalocele and testicular hydroceles.

The diagnosis of BWS was established based on the presence of three major symptoms including hemihyperplasia, macroglossia, and omphalocele and more than two minor symptoms, including prematurity, hypoglycemia, and characteristic facial features such as hypoplasia of the middle part of the face, hypertelorism, and a prominent occiput [1,19].

Blood samples were collected from the newborn for genetic testing using the MLPA kit ME030-BWS/SRS by MRC-Holland to check for Beckwith–Wiedemann Syndrome and Silver–Russell Syndrome. The genetic examination showed a normal methylation pattern in the 11p15.5 region (genes: *H19*, *IGF2*, *KCNQ1*, *CDKN1C*), and no rsa (*ME030-BWS/SRS*) × 2 deletion/duplication was found. Additionally, a section of genomic DNA covering the coding fragments of the *CDKN1C* gene (exons 1 and 2) along with the surrounding intronic sequences was sequenced using the Sanger method, but no pathogenic variants were identified.

The neonate’s chromosomal analysis revealed a normal male karyotype: 46, XY using GTG-450 banding technique. Serial measurements of alpha-fetoprotein (AFP) were taken postpartum, with levels recorded as 3644.5 ng/mL at 1.5 months and 767 ng/mL at 2.5 months.

The newborn is currently developing normally and is under the constant supervision of the endocrinology clinic for children and the genetics clinic. As per the recommendations, he is scheduled to undergo abdominal ultrasound and AFP testing every 3 months, along with other necessary tests. The child also underwent an anthropological consultation at the age of 6 and 12 months. The results of the consultation are presented in Table 3.

## 3. Discussion

Beckwith–Wiedemann Syndrome is a rare genetic disorder associated with alterations in the imprinted domain of chromosome 11p15.5. BWS manifests a wide range of clinical features, including neonatal hypoglycemia, macroglossia, abdominal wall defects, and a heightened risk of embryonal tumors.

A significant proportion of BWS patients exhibit the biallelic expression of the insulin-like growth factor 2 (*IGF2*) gene and an epigenetic mutation leading to loss of imprinting of the *KCNQ1OT1* transcript. Both *IGF2* and *KCNQ1OT1* reside in the two imprinted domains that appear to be developmentally dysregulated in BWS. The gene also encodes the cyclin-dependent kinase inhibitor *p57KIP2*. Abnormalities in this gene and its imprinting status have also been linked to BWS and the associated risk of embryonal tumors [20,21].

A notable characteristic of BWS is lateralized overgrowth. This asymmetry is a key clinical feature and can be correlated with various molecular anomalies within the imprinted 11p15.5 region. This case report presents a unique scenario where BWS was identified in a newborn born to a mother diagnosed with HELLP syndrome and preeclampsia, and the neonate exhibited fetal growth restriction (FGR) during pregnancy. To our knowledge, in all prior reports, in case of FGR and preeclampsia or HELLP developing in the mother of a newborn with BWS, a typical pathogenic genetic variation in the most typical genes was always reported. Notably, genetic testing revealed an absence of *CDKN1C* typical pathogenic genetic variation, which is commonly associated with BWS [1,2,3,4,5,6,7,8].

The association between BWS and maternal complications such as HELLP syndrome and preeclampsia has been rarely reported in the literature [10,22,23]. The absence of *CDKN1C* mutations in this case is an intriguing finding. *CDKN1C*, encoding the cyclin-dependent kinase inhibitor 1C (*p57KIP2*), is one of the key genes involved in the pathogenesis of BWS. Aberrations in *CDKN1C* methylation or mutations in the gene have been well documented in classic BWS cases. The lack of *CDKN1C* mutations in our case suggests the involvement of alternative genetic mechanisms in the development of BWS [24,25]. It is possible that other imprinted genes or epigenetic changes within the 11p15.5 region are responsible for the observed BWS phenotype. It is plausible that the underlying genetic basis could be identified in more advanced genetic investigations, including whole-genome sequencing, which is not routinely available in the clinical setting. RNA sequencing also offers a comprehensive approach to understanding the molecular underpinnings of BWS. As BWS is associated with abnormalities in the imprinting of genes on chromosome 11p15.5, RNA-seq can be used to profile the expression of genes and can detect alternative splicing events that might be unique to BWS, providing insights into the disease’s molecular mechanisms [1,26,27].

In conclusion, this case report describes a rare scenario of BWS in a newborn born to a mother with HELLP syndrome and preeclampsia, presenting with FGR and an absence of *CDKN1C* mutations. The absence of *CDKN1C* mutations shows that it does not play a key role in the development of HELLP Syndrome and other complications in pregnancy. It highlights the need for considering alternative genetic etiologies when *CDKN1C* mutations are not detected in BWS cases, especially since, according to the literature, approximately 20% of BWS cases are not related to *CDKN1C* mutations [1]. Further research should focus on identifying the underlying genetic basis of BWS in cases without *CDKN1C* mutations and on exploring the potential association between BWS and placental dysfunction.

## 4. Conclusions

This case report highlights the complexity of BWS diagnosis. The presence of HELLP syndrome, preeclampsia, and FGR in the newborn adds to the complexity of clinical management and warrants further investigation into the underlying genetic mechanisms.

## Figures and Tables

**Table 1 ijms-24-13360-t001:** Fetal ultrasound examinations.

	GA (Weeks + Days)									
Parameters	20 + 2	23 + 1	27 + 0	30 + 3	31 + 2	32 + 4	33 + 3	34 + 4	35 + 0	36 + 1
EFW (g)	328	541	928	1212	1301	1421	1490	1599	1732	1760
EFW (Percentile)	22.2	20.5	8.0	0	0	0	0	0	0	0
CPR (Percentile)	-	-	47	41	39	44	37	34	35	26
Mean PI UtA (Percentile)	62	56	48	28	29	22	34	33	37	40
STV (ms)	-	-	-	-	-	-	-	10.5	8.9	11.3
MVP (cm)	5.2	3.3	5.7	3.7	4.2	5.6	5.2	3.9	4.1	5.2

Abbreviations: CPR—cerebroplacental ratio; EFW—estimated fetal weight; MVP—maximum vertical pocket; PI—pulsatility index; STV—short-term variation; UtA—uterine artery.

**Table 2 ijms-24-13360-t002:** Laboratory results of the patient (mother) before and after delivery.

	Hospitalization Day to/after Birth (Delivery at Day 0)								
Parameters	−2	0	1	2	3	4	5	6	7
Leukocytes (thous/µL)	7.8	17.0	11.6	13.5	13.8	12.3	12.0	9.3	9.4
Erythrocytes (mln/µL)	3.69	3.32	3.10	2.92	3.20	2.94	3.08	3.27	3.53
Hemoglobin (g/dL)	12.7	11.5	10.6	9.9	11.1	10.5	10.5	11.3	12.3
Hematocrit (%)	37.9	32.9	31.0	29.6	32.8	30.5	31.8	34.0	36.8
PLT (10^9^/L)	177	44	24	47	106	162	222	299	411
AST (U/L)	21.3	-	254.2	94.5	129.3	58.6	28.9	23.1	23.0
ALT (U/L)	12.8	-	377.3	254.7	277.7	197.5	133.3	105.3	81.9
Creatinine (mg/dL)	0.72	-	0.67	0.55	0.64	-	-	0.61	0.62
D-Dimer (µg/mL)	-	2.61	5.39	20.99	25.38	5.85	1.72	1.09	0.92
LDH (U/L)	-	-	951.0	709.0	-	-	-	-	-
CRP (mg/L)	-	-	77.48	45.5	-	-	-	3.18	2.6

**Table 3 ijms-24-13360-t003:** Anthropological measurements.

Parameters	6 Months Old	12 Months Old
Body length (mm)	488 (−4.7 SD)	640 (−2.2 SD)
Body weight (kg)	3.69 (+36%)	7.25 (+4%)
Length of the upper limbs (mm)	185 (−2.6 SD)	240 (−1.5 SD)
Length of the lower limbs (mm)	193 (−2.6 SD)	245 (−2.5 SD)
Head circumference (mm)	362 (−3.1 SD)	428 (−1.3 SD)
Thorax circumference (mm)	330 (−4.3 SD)	415 (−1.3 SD)
Body mass index	15.49	17.7

## Data Availability

Further data on patients included in this study can be made available on request to the corresponding author.

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
