# Peer review of "Beckwith–Wiedemann Syndrome in Newborn of Mother with HELLP Syndrome/Preeclampsia: An Analysis of Literature and Case Report with Fetal Growth Restriction and Absence of CDKN1C Typical Pathogenic Genetic Variation"

_ijms, 2023, doi:10.3390/ijms241713360_

Round 1

Reviewer 1 Report

Dear authors, thank you for very interesting case report,

I have several comments and first one - based on given description have doubts about BWS diagnose  in line 155 is mentioned major and minor symptoms, of which hemi hyperplasia, omphalocele hypoplasia is not described in section 3.2. so it raises a questions. the same as hypoplasia of the middle part of the face is not included in criteria (https://doi.org/10.1038/nrendo.2017.166 ). The doubts are even more significant as molecular testing were negative although according to diagnostic algorithm mosaicism were not excluded etc. It should be better desribed symptoms of the patient that is leading to BWS or probably its is BWS spectrum disease, but in form as it is it gives impression about wrong first diagnose. 
The rest is only small comments:
- in line 50 instead of uncommon better to use rare
- line 57 in genetics term multifactorial means smth completely different so please avoid of using it, from the context it seems that better term would be pleiotropic.
- line 62 and other places - gene names should be in italyc
in line 73, and 188 - used term mutation, that is incorrect - "typical pathogenic genetic variation" according to ACMG and HGVS would be better
- line 133 table 2 doesn't have reference in the text but although from the given title it is not clear who is the patient described - mother or newborn;
- in discussion as well in introduction in clinical descriptions is stressed overgrowth, embryonical tumours that is not present in the case, but no word about asymmetry, although that is mentioned in used reference (https://doi.org/10.1038/nrendo.2017.166 ).

Author Response

Dear Reviewer,

Thank you for your thoughtful comments and insights on our case report. We appreciate the time and effort you have taken to review our manuscript, and we acknowledge the areas of concern you have highlighted. Please find our responses to your comments below:

1. Concerns about BWS Diagnosis: We understand your concerns regarding the diagnosis of BWS based on the described symptoms. Postnatally, the only symptom indicative of BWS observed was macroglossia. As the child aged over the subsequent weeks, additional symptoms began to manifest. This led to an initial (3 weeks after birth) genetic consultation at the University Children's Hospital in Krakow, Poland, which serves as the reference Clinical Genetics Center for children. The diagnosis of BWS was established based on the presence of three major symptoms (hemihyperplasia, macroglossia, and omphalocele) and more than two minor symptoms, including prematurity, hypoglycemia, and characteristic facial features such as hypoplasia of the middle part of the face, hypertelorism, and a prominent occiput. We appreciate your attention to this detail. The text has been revised accordingly, and an appropriate reference has been incorporated. A panel consisting of Clinical Genetics specialists, pediatricians, and anthropologists unanimously confirmed the diagnosis of BWS.

2. Specific Comments:

Line 50: We will replace "uncommon" with "rare" as suggested.
Line 57: We acknowledge the misuse of the term "multifactorial." We will replace it with "pleiotropic" to better reflect the intended meaning.
Line 62 and other instances: We will ensure that gene names are italicized throughout the manuscript.
Lines 73 and 188: We will replace the term "mutation" with "typical pathogenic genetic variation" in accordance with ACMG and HGVS guidelines.
Line 133: We will provide a reference for Table 2 in the text and clarify the title to specify whether the patient described is the mother or the newborn.

3. Discussion and Introduction: We will revise these sections to address the emphasis on overgrowth, which were not present in our case. We will also ensure that the aspect of asymmetry, as mentioned in the provided reference, is adequately discussed.

Once again, thank you for your valuable feedback. We believe that addressing these comments will significantly improve the quality and clarity of our manuscript. We are committed to making the necessary revisions to ensure the accuracy and comprehensiveness of our report.

Warm regards,

Jakub Staniczek

Reviewer 2 Report

The authors well describe an interesting case of a BWS without CDKN1C mutations and born from a woman with a history of preeclampsia and HELLP syndrome.

Minor revision:

-Please correct CDKN1C gene in italics. Check the whole text.

- At line 62-63, the authors talk about CDKN1C implications. The same at lines 195-196. Please, add references regarding CDKN1C, i.e.:

Stampone E, Caldarelli I, Zullo A, Bencivenga D, Mancini FP, Della Ragione F, Borriello A. Genetic and Epigenetic Control of CDKN1C Expression: Importance in Cell Commitment and Differentiation, Tissue Homeostasis and Human Diseases. Int J Mol Sci. 2018 Apr 2;19(4):1055. doi: 10.3390/ijms19041055. PMID: 29614816; PMCID: PMC5979523.

Stampone E, Bencivenga D, Barone C, Di Finizio M, Della Ragione F, Borriello A. A Beckwith-Wiedemann-Associated CDKN1CMutation Allows the Identification of a Novel Nuclear Localization Signal in Human p57Kip2. Int J Mol Sci. 2021 Jul 11;22(14):7428. doi: 10.3390/ijms22147428. PMID: 34299047; PMCID: PMC8305445.

The authors of these works, deeply described the genetic and epigenetic mechanisms that control CDKN1C expression and the alterations that cause three human hereditary syndromes, characterized by altered growth rate, like BWS. Regarding this latter, they analyzed the effect of a BWS mutation identifying a novel nuclear localization signal of the human protein p57Kip2.

-line 72. Please use the abbreviation BWS. Check in the whole text.

-lines 200-201. Please consider also RNA seq. 

-lines 214-215. Singolar or plural? Please, check the whole text for grammar errors.

Check grammar errors. Minor revision requested

Author Response

Dear Reviewer,

Thank you for your valuable feedback. We have carefully addressed each of your comments and made the necessary corrections to our manuscript. Below is a detailed response to each of your points:

1. Italicization of CDKN1C gene: We have reviewed the entire manuscript and ensured that the gene name "CDKN1C" is consistently italicized throughout the text.

2. References regarding CDKN1C at lines 62-63 and 195-196: We appreciate your suggestion. We have added the following references to provide more context on the implications of CDKN1C:

  • Stampone E, Caldarelli I, Zullo A, et al. Genetic and Epigenetic Control of CDKN1C Expression: Importance in Cell Commitment and Differentiation, Tissue Homeostasis and Human Diseases. Int J Mol Sci. 2018 Apr 2;19(4):1055. doi: 10.3390/ijms19041055. PMID: 29614816; PMCID: PMC5979523.
  • Stampone E, Bencivenga D, Barone C, et al. A Beckwith-Wiedemann-Associated CDKN1C Mutation Allows the Identification of a Novel Nuclear Localization Signal in Human p57Kip2. Int J Mol Sci. 2021 Jul 11;22(14):7428. doi: 10.3390/ijms22147428. PMID: 34299047; PMCID: PMC8305445.

3. Abbreviation of BWS at line 72: We have replaced the full term "Beckwith-Wiedemann Syndrome" with its abbreviation "BWS" at line 72 and ensured its consistent use throughout the manuscript.

4. Consideration of RNA seq at lines 200-201: We have incorporated the mention of RNA sequencing (RNA seq) at the specified lines to provide a more comprehensive view of the methodologies.

4. Grammar check at lines 214-215 and throughout the manuscript: We have thoroughly reviewed lines 214-215 and made the necessary grammatical corrections. Additionally, we have conducted a comprehensive grammar check throughout the manuscript to ensure clarity and accuracy.

We hope that these revisions address your concerns, and we believe that they have enhanced the quality of our manuscript. We appreciate your time and effort in reviewing our work and look forward to your feedback.

Warm regards,

Jakub Staniczek